# Disentangled Concepts Speak Louder Than Words: Explainable Video Action Recognition

**Jongseo Lee**[1]    **Wooil Lee**[1]    **Gyeong-Moon Park**[2†]    **Seong Tae Kim**[1†]    **Jinwoo Choi**[1†]

[1]Kyung Hee University, Republic of Korea
[2]Korea University, Republic of Korea

{jong980812, lwi2765, st.kim, jinwoochoi}@khu.ac.kr, gm-park@korea.ac.kr

## Abstract

Effective explanations of video action recognition models should *disentangle* how movements unfold over time from the surrounding spatial context. However, existing methods—based on saliency—produce entangled explanations, making it unclear whether predictions rely on motion or spatial context. Language-based approaches offer structure but often fail to explain motions due to their tacit nature—intuitively understood but difficult to verbalize. To address these challenges, we propose Disentangled Action aNd Context concept-based Explainable (`DANCE`) video action recognition, a framework that predicts actions through disentangled concept types: motion dynamics, objects, and scenes. We define motion dynamics concepts as human pose sequences. We employ a large language model to automatically extract object and scene concepts. Built on an ante-hoc concept bottleneck design, `DANCE` enforces prediction through these concepts. Experiments on four datasets—KTH, Penn Action, HAA500, and UCF-101—demonstrate that `DANCE` significantly improves explanation clarity with competitive performance. We validate the superior interpretability of `DANCE` through a user study. Experimental results also show that `DANCE` is beneficial for model debugging, editing, and failure analysis. Our project page is available at https://jong980812.github.io/DANCE/

## 1   Introduction

Recent advances in video action recognition [56, 11, 28, 13, 32, 53, 30, 2] have led to impressive performance across diverse benchmarks. To deploy such high-performing video models in real-world applications, explaining their predictions becomes essential for ensuring trust, transparency, and accountability [10, 20]. Despite this need, the decision-making processes of video action recognition models remain largely opaque, and systematic approaches to explanation are still underexplored.

From a cognitive science perspective, humans interpret complex information more effectively when it is presented in a *structured* format—i.e., broken down into meaningful and separable components [36, 34, 35, 51]. Interpretability further improves when each component is expressed in a clear and unambiguous manner [10, 55, 33]. Notably, humans perceive actions by separately analyzing two distinct factors: (i) how movements evolve over time (temporal dynamics) and (ii) what physical context surrounds those movements, such as objects and scenes (spatial context) [17, 16, 27]. Therefore, to align with human reasoning process, a video explainable AI (video XAI) should explain a model's prediction in a *structured* way—explicitly *disentangling* and attributing its decisions to temporal dynamics and spatial context.

Meanwhile, existing approaches in video XAI largely follow two strategies: extending image-based feature attribution methods [15, 23, 42, 14, 46] across the time axis [50, 18, 31] or clustering spatio-temporal tubelets to discover high-level concepts [26, 21, 43]. However, these methods do not

---

[†]Corresponding authors.

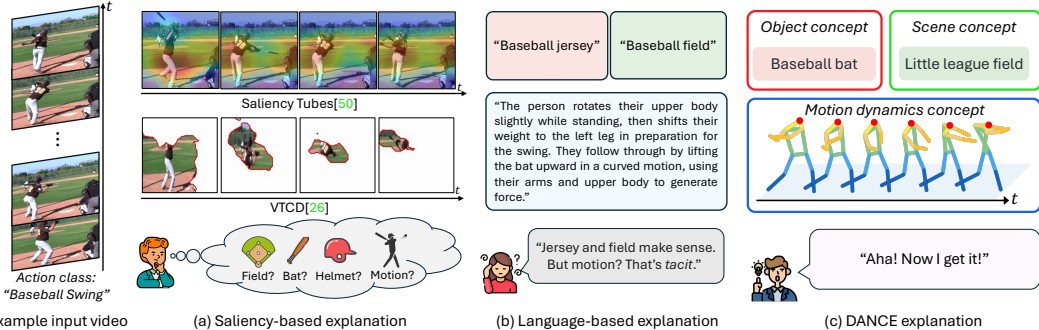

Figure 1: **Disentangled concepts speak louder than words.** Spatio-temporal attribution methods provide *unstructured* explanations that are often ambiguous to human users. Given a *Baseball Swing* video, (a) visual explanations from 3D-saliency [50] and VTCD [26] fail to clarify whether the prediction is driven by motion (e.g., torso twist), objects (e.g., jersey/helmet), or scene context (e.g., baseball field). (b) Language-based approaches offer more structure but remain ambiguous for motion, as it is *tacit knowledge*—intuitively understood but hard to verbalize. Verbal descriptions of motion often lack clarity and are difficult to interpret. (c) In contrast, DANCE *disentangles* motion and context to provide *structured* explanations. Pose sequences capture motion dynamics in an intuitive, appearance-invariant form, while we clearly convey object and scene concepts via text.

disentangle temporal dynamics from spatial context in their explanations. Instead, they highlight localized regions of input videos in an unstructured and entangled manner—making it difficult to attribute predictions to specific types of evidence. For example, in Figure 1 (a), for the action *Baseball Swing*, it remains unclear whether the model's prediction is based on the twisting motion of the torso, the appearance of the player (e.g., the jersey), or the scene (e.g., baseball field).

A potential direction for structured explanations is to use language-based approaches [38, 60, 39, 37, 1]. Language-based approaches could provide structured and human-readable text descriptions, as illustrated in Figure 1 (b). While these methods can effectively capture spatial context or high-level semantics, they often struggle to express *motion dynamics* clearly. This challenge arises because motion dynamics often fall under the category of *tacit knowledge*—knowledge that is intuitively understood and applied, but difficult to verbalize or explain explicitly [41]. As shown in Figure 1 (b), verbally describing the swinging motion of a torso is nontrivial, and even when verbalized, such descriptions tend to be overly verbose and cognitively difficult for users to interpret.

To address the challenges of structured and motion-aware explanation, we propose Disentangled Action aNd Context concept-based Explainable (DANCE) video action recognition framework. As illustrated in Figure 1 (c), DANCE provides explanations based on three *disentangled* concept types: (i) motion dynamics, (ii) object, and (iii) scene. To capture fine-grained temporal patterns, we define motion dynamics concepts as human pose sequences. These pose sequences offer an appearance-agnostic representation of motion, enabling users to intuitively understand how an action unfolds over time without being distracted by irrelevant visual factors such as clothing or background. In parallel, we define object and scene concepts as action-related elements extracted using a large language model, allowing a model to incorporate spatial context concepts without manual annotation.

To ensure inherent explainability, DANCE adopts an ante-hoc design based on the concept bottleneck framework [25]. We insert a concept layer between the backbone and the final classifier, enforcing the model to first predict concept activations before predicting the final action label. The concept layer comprises nodes for motion dynamics, object, and scene concepts. This *disentangled* design ensures that action predictions are explicitly grounded in both dynamic (pose sequence-based) and static (object and scene) concept types. As a result, explanations produced by DANCE are not only faithful to the model's reasoning but also well-aligned with human cognitive mechanisms.

To validate the effectiveness of DANCE, we conduct experiments on four video action recognition datasets: KTH [45], Penn Action [61], HAA500 [9], and UCF-101 [49]. Results show that DANCE significantly enhances explanation clarity by disentangling motion dynamics and spatial context, while maintaining competitive recognition performance against a model without interpretability. A user study further demonstrates that explanations generated by DANCE are more faithful and interpretable compared to those of prior approaches. Extensive qualitative comparisons also highlight the superior structure and transparency of DANCE's explanations. Finally, we showcase the practical utility of DANCE across several downstream tasks—including model debugging, editing, and failure

case analysis —underscoring its broader potential for understanding and improving video recognition models.

We summarize our major contributions as follows:

- We propose DANCE, a novel video XAI framework that provides *structured* and *motion-aware* explanations by *disentangling* motion dynamics and spatial context concepts.
- We introduce a label-free pipeline that *automatically discovers* motion dynamics concepts via clustering of human pose sequences and spatial context concepts through LLM querying. The proposed pipeline allows DANCE to capture fine-grained motion patterns, objects, and scenes without manual annotations.
- We conduct comprehensive evaluations across four datasets, assessing both explainability and performance through a user study, qualitative comparisons, and ablation studies. We further demonstrate the practical utility of DANCE in model debugging and editing.

## 2 Related Work

**Video action recognition.** The video action recognition task is classifying human actions from a temporally trimmed input video. Early approaches, such as two-stream CNNs [48], 3D CNNs [54, 12, 13], and temporal shift modules [32], jointly encode spatial and temporal features to capture motion and context. More recently, transformer-based architectures [3, 40, 11, 53, 28, 2] have achieved substantial performance gains, largely due to large-scale pretraining. Despite these advances, the decision-making processes of most video action recognition models remain opaque, as predictions are made through complex, non-interpretable feature interactions. In this work, we propose DANCE that predicts actions based on *disentangled*, human-interpretable concepts, thus making the model's reasoning process more transparent and well-aligned with human cognition.

**Explainable video action recognition.** Explaining the decision-making process of video action recognition models remains a relatively under-explored area. We can categorize existing methods into post-hoc explanation approaches—such as feature attribution [50, 31, 18]—and concept discovery methods [21, 26, 43]. These approaches typically use attribution or optimization techniques to identify input regions (e.g., pixels or spatio-temporal tubelets) that contribute most to the model's prediction. As a result, the explanations they produce are often *unstructured and entangled*, making it difficult to attribute predictions to distinct types of reasoning, such as motion dynamics versus spatial context. Moreover, optimization-based methods [21, 26, 43] require additional computational cost, as they need an optimization phase every time for generating an explanation. In contrast, we propose an ante-hoc framework that explicitly *disentangles* temporal dynamics and spatial context concepts to produce structured, human-aligned explanations. Because DANCE is inherently explainable by design, it can generate interpretable explanations in a single forward pass, without a post-hoc optimization.

**Disentangled/Decomposed explanations.** Recently, there have been efforts to provide explanations disentangled into interpretable components in the image domain [62, 24, 4, 8, 25, 38, 60, 39, 47]. Concept bottleneck models (CBMs) insert a concept layer between the backbone and the final classifier, forcing predictions to be made explicitly through disentangled human-interpretable concepts [25, 38, 60, 39, 47]. Other approaches disentangle what concepts influence the prediction and where the concepts occurs [52, 1, 4]. A separate line of work addresses the limitations of attribution-based explanations—entangled attribution maps—by disentangling intermediate-layer representations into concept subspaces [8] or selecting a compact set of informative attribution regions [5]. While we also aim to provide disentangled explanations, our work differs in that we tackle the under-explored problem of explainable video action recognition. To the best of our knowledge, DANCE is a pioneering work in video XAI by explicitly *disentangling* motion dynamics and spatial context concepts, enabling structured, interpretable, and human-aligned explanations for video model decisions.

## 3 DANCE

We introduce DANCE, an explainable video action recognition framework that produces *structured* and *motion-aware* explanations for its predictions. As illustrated in Figure 2, DANCE explains each prediction using three types of *disentangled* concepts: (i) motion dynamics, (ii) objects, and (iii) scenes. To capture fine-grained temporal patterns, we define motion dynamics concepts as representative human pose sequences extracted from training videos. These pose sequences offer appearance-agnostic representations of temporal motion, allowing users to clearly understand how an action unfolds over time—without being distracted by visual factors such as clothing, objects, or

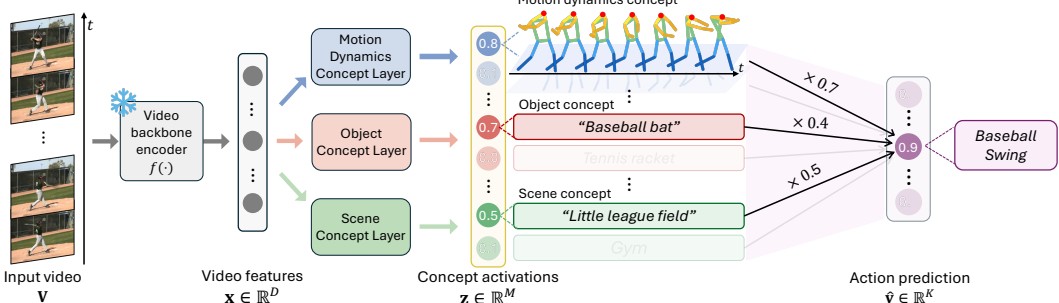

Figure 2: **Overview of DANCE.** Given an input video, DANCE first extracts video features using a pretrained video backbone encoder. Then, three disentangled concept layers project the video features onto their own concept space-motion dynamics, object, and scene-producing disentangled activations. The interpretable classification layer linearly combines these concept activations to predict the action class. By explicitly disentangling concept types, DANCE provides structured explanations that better align with how humans perceive actions by separating motion dynamics from the spatial context.

background. In parallel, we use a large language model (LLM) to extract spatial context concepts, identifying relevant objects and scenes associated with each action.

To ensure a transparent prediction, we adopt an ante-hoc design based on the concept bottleneck framework [25, 38]. As shown in Figure 2, we insert a concept layer between the backbone and the final classifier. Given a video, DANCE first predicts the activation of disentangled concepts, then uses these activations to produce the final action prediction. Through our disentangled concept bottleneck design, we ensure that an explanation from DANCE is *structured* and *motion-aware*. The remainder of this section is organized as follows. We introduce the concept bottleneck architecture in Section 3.1, detail our concept discovery process in Section 3.2, present training procedures in Section 3.3.

## 3.1 Preliminary: Concept Bottleneck Model

Let us denote a training dataset as $D = (\mathbf{V}_i, \mathbf{c}_i, \mathbf{y}_i)_{i=1}^N$, where $\mathbf{V}_i$ is the $i$-th input video, $\mathbf{c}_i \in \{0,1\}^M$ is a binary vector indicating the presence of $M$ concepts, and $\mathbf{y}_i \in \{0,1\}^K$ is a one-hot vector indicating the ground-truth action label among $K$ classes. Given a training sample $(\mathbf{V}_i, \mathbf{c}_i, \mathbf{y}_i)$, we first extract a $D$-dimensional video-level feature vector $\mathbf{x}_i = f(\mathbf{V}_i) \in \mathbb{R}^D$ using a pre-trained video backbone encoder $f(\cdot)$. We then project $\mathbf{x}_i$ into $M$ concept activations using a linear concept layer $g(\cdot; \mathbf{W}_C)$ parameterized by weights $\mathbf{W}_C \in \mathbb{R}^{M \times D}$: $\mathbf{z}_i = g(\mathbf{x}_i; \mathbf{W}_C) \in \mathbb{R}^M$. Then, a linear classifier $h(\cdot; \mathbf{W}_A)$ with a softmax activation predicts an action label based on the concept activations: $\hat{\mathbf{y}}_i = h(\mathbf{z}_i; \mathbf{W}_A) \in \mathbb{R}^K$.

Unlike prior works [25, 38], DANCE explicitly disentangles three types of concepts—motion dynamics, objects, and scenes—to provide *structured* and more intuitive explanations. To achieve this, we partition the parameters of the concept layer $\mathbf{W}_C$ into three disjoint parameters: $\mathbf{W}_C = [\mathbf{W}_C^m; \mathbf{W}_C^o; \mathbf{W}_C^s]$, where $\mathbf{W}_C^m \in \mathbb{R}^{M_m \times D}$, $\mathbf{W}_C^o \in \mathbb{R}^{M_o \times D}$, and $\mathbf{W}_C^s \in \mathbb{R}^{M_s \times D}$ correspond to the parameters for motion dynamics, object and scene concepts, respectively. Here, $M_m$, $M_o$, and $M_s$ denote the number of motion dynamics, object and scene concepts, respectively. We represent motion dynamics concepts using *2D human pose sequences*, which explicitly capture how the human body moves over time in an appearance-agnostic manner. For both object and scene concepts, we use intuitive *text descriptions*, e.g., *baseball bat*, *tennis court*, that reflect the spatial context associated with each action.

## 3.2 Concept Discovery

For each concept type, we first discover a representative set of concepts using only the training videos from the target dataset. We then automatically annotate each video with the presence or absence of these concepts, without requiring any *human supervision*.

### 3.2.1 Motion Dynamics Concept

As shown in Figure 3 (a), we extract 2D pose sequences from all training videos and apply clustering to discover representative motion dynamics concepts. This allows us to build a compact and interpretable vocabulary of movement patterns.

**Key clip selection.** In video sequences, not all clips are equally informative [22, 7, 6]; only a few temporally localized segments—such as *wind-up*, *stride*, or *release* in a baseball pitch—contain

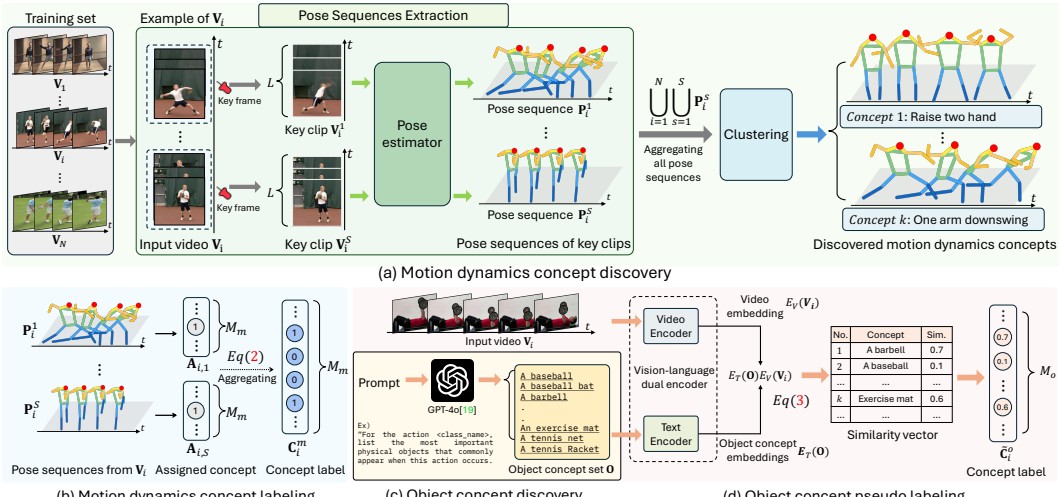

Figure 3: **Concept discovery and labeling process of DANCE.** (a) Given a training video, we extract $S$ key clips with length $L$ centered at keyframes identified by a keyframe detection algorithm. We then apply a 2D pose estimator to obtain human pose sequences from these key clips. By clustering all pose sequences across the training set, we cluster them to define each cluster as a motion dynamics concept. (b) For each video, we derive binary motion dynamics concept labels by aggregating the cluster assignment tensor across its key clips. (c) To discover object concepts, we query GPT-4o [19] with prompts containing action class names, yielding a set of object concepts for the dataset. (d) Given a video and the object concept set, we compute concept pseudo labels using a vision-language dual encoder. Specifically, we obtain a concept pseudo label vector by multiplying the object concept embedding matrix with the video embedding vector. We can obtain scene concept labels analogously.

distinctive motion cues critical for action recognition. To focus on such informative segments, we extract pose sequences from *key clips* only. We first detect keyframes by running an off-the-shelf method [*] using pixel value differences. For each selected keyframe, we extract a short video clip $\mathbf{V}_i^s$ of fixed length $L$ centered at that frame, where $s \in \{1, \cdots, S\}$ is the key clip index. This targeted sampling strategy allows us to concentrate the concept discovery process on *primitive*, discriminative motion patterns that are frequently *shared* across different instances of the same action class—and in some cases, across classes. Please refer to the supplementary materials.

**Pose sequence extraction.** For each key clip $\mathbf{V}_i^s$, we apply a 2D pose estimation model [59] to every frame to obtain a pose sequence $\mathbf{P}_i^s \in \mathbb{R}^{L \times J \times 2}$, where $J$ is the number of joints. To ensure high-quality motion dynamics representations, we filter out pose sequences with low average joint confidence or large discontinuities in joint coordinates between consecutive frames. For further implementation details, please refer to the supplementary materials.

**Concept discovery.** To discover motion dynamics concepts, we first aggregate all pose sequences from the training videos into a unified set: $\mathbb{P} = \bigcup_{i=1}^{N} \bigcup_{s=1}^{S} \mathbf{P}_i^s$, where $\mathbf{P}_i^s$ denotes the pose sequence from the $s$-th key clip of the $i$-th video. To group similar motion patterns, we apply a clustering algorithm, e.g., FINCH [44], to the aggregated set $\mathbb{P}$, as illustrated in Figure 3 (c). We flatten each pose sequence into a feature vector before clustering. We define each resulting cluster as a distinct *motion dynamics concept* and assign it a unique concept index $k \in \{1, \ldots, M_m\}$, where $M_m$ is the total number of motion dynamics concepts. Based on the clustering results, we construct a binary cluster assignment tensor $\mathbf{A} = [a_{i,s,k}] \in \{0,1\}^{N \times S \times M_m}$, where each element $a_{i,s,k}$ is defined as:

$$a_{i,s,k} = \begin{cases} 1, & \text{if } \mathbf{P}_i^s \text{ belongs to cluster } k, \\ 0, & \text{otherwise.} \end{cases} \tag{1}$$

**Concept labeling.** For each training video $\mathbf{V}_i$, we assign motion dynamics concept labels by checking whether any of its pose sequences $\mathbf{P}_i$ belong to a given cluster $k$:

$$\mathbf{c}_{i,k}^m = \mathbb{I}(\sum_{s=1}^{S} a_{i,s,k}), \tag{2}$$

---

[*]https://github.com/joelibaceta/video-keyframe-detector

where $\mathbb{I}(\cdot)$ is the indicator function. As a result, we obtain a binary motion dynamics concept label vector $\mathbf{c}_i^m \in \{0,1\}^{M_m}$ indicating which motion dynamics concepts are present in video $\mathbf{V}_i$. Note that this labeling process is entirely *unsupervised*, requiring no manual annotations.

### 3.2.2 Object and Scene Concept

**Concept Discovery.** To discover intuitive and clear concepts in an unsupervised manner, we leverage a large language model, as illustrated in Figure 3 (d). For each action class, we query GPT-4o [19] with two prompts: i) *"For the <action class>, list the most important physical objects that commonly appear when this action occurs."* and ii) *"List the most common places or background scenes where <action class> typically occurs. Do not include objects or equipment"*. These prompts yield a diverse and semantically meaningful set of candidate object and scene concepts associated with each action. To improve concept quality and reduce redundancy, we follow prior work [38] by applying post-processing filters—removing overly long phrases, near-duplicates, and concepts that are overly similar to the action class name. For more implementation details and examples, please refer to the supplementary materials.

**Concept Pseudo Labeling.** To avoid manual concept annotation, we employ a vision-language dual encoder [57] to generate concept pseudo labels for each training video $\mathbf{V}_i$. Let $E_V(\cdot)$ and $E_T(\cdot)$ denote the video and text encoders of the dual encoder, respectively. We first obtain a video embedding vector $E_V(\mathbf{V}_i) \in \mathbb{R}^D$, where $D$ is the shared embedding dimension. We then encode the object concept set $\mathbf{O}$ obtained from GPT-4o using the text encoder to obtain an object concept embedding matrix $E_T(\mathbf{O}) \in \mathbb{R}^{M_o \times D}$, where $M_o$ denotes the number of object concepts. Given object concept embeddings $E_T(\mathbf{O})$ and the video embedding $E_V(\mathbf{V}_i)$, we compute the object pseudo concept label vector $\tilde{\mathbf{c}}_i^o$ as:

$$\tilde{\mathbf{c}}_i^o = E_T(\mathbf{O})E_V(\mathbf{V}_i) \in [0,1]^{M_o}. \tag{3}$$

Note that $\tilde{\mathbf{c}}_i^o$ is a soft label. We can obtain scene concept labels $\tilde{\mathbf{c}}_i^s \in [0,1]^{M_s}$ analogously using the scene concept embedding matrix $E_T(\mathbf{S}) \in \mathbb{R}^{M_s \times D}$ and the video embedding $E_V(\mathbf{V}_i)$. For more details, please refer to the supplementary materials.

### 3.3 Training

We freeze the pretrained video backbone encoder $f(\cdot)$ and train the concept layer $g(\cdot; \mathbf{W}_C)$ and the final classification layer $h(\cdot; \mathbf{W}_A)$ in two separate stages, using concept labels derived in Section 3.2. For brevity, we omit the batch dimension and the sample index $i$ in the following descriptions.

**Motion dynamics concept layer.** Since the motion dynamics label $\mathbf{c}_i^m \in \{0,1\}^{M_m}$, derived from (2), is a multi-label binary vector rather than a one-hot vector, we train the motion dynamics concept parameters using the binary cross-entropy loss. Given the motion dynamics concept activations $\mathbf{z}^m = g(\mathbf{x}; \mathbf{W}_C^m)$, we apply the sigmoid activation $\sigma(\cdot)$ to each element and define the loss as:

$$L_m = -\frac{1}{M_m} \sum_{k=1}^{M_m} [c_k^m \log(\sigma_k(\mathbf{z}^m)) + (1 - c_k^m) \log(1 - \sigma_k(\mathbf{z}^m))]. \tag{4}$$

Here, $\sigma_k(\mathbf{z}^m)$ denotes the $k$-th element of $\sigma(\mathbf{z}^m)$.

**Object and scene concept layer.** To train the object concept layer, we follow prior work [38] and apply the cosine cubed loss between the pseudo label vector $\tilde{\mathbf{c}}^o$ (derived from (3)) and the object concept activations $\mathbf{z}^o = g(\mathbf{x}; \mathbf{W}_C^o)$. The cosine cubed loss emphasizes directional alignment between the predicted and target concept representations while being invariant to scale. For additional details, we refer readers to the methodology described in [38]. We train the scene concept layer in an analogous manner using the corresponding pseudo labels and activations.

$$L_o = -\frac{\mathbf{z}^{o3} \cdot \tilde{\mathbf{c}}^{o3}}{\left\|\mathbf{z}^{o3}\right\|_2 \cdot \left\|\tilde{\mathbf{c}}^{o3}\right\|_2}. \tag{5}$$

**Interpretable classifier.** We *freeze* the learned concept layers and train a final linear classifier that predicts actions solely based on the concept activations. By enforcing predictions to be made through disentangled and interpretable concepts, we promote transparent and structured explanations. Let $\mathbf{z} = [\mathbf{z}^m; \mathbf{z}^o; \mathbf{z}^s]$ denote the concatenated motion dynamics, object, and scene concept activations. We train the classifier using the standard cross-entropy loss between the ground-truth action label $\mathbf{y}$ and

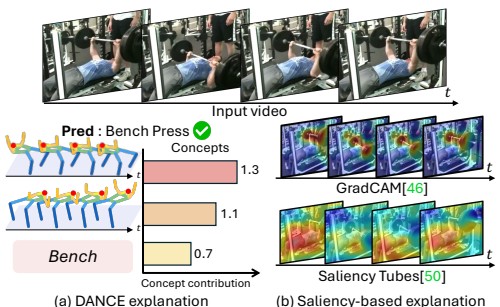

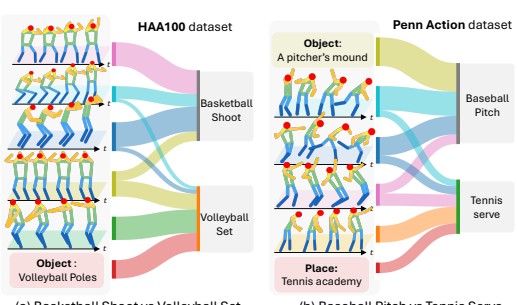

Figure 4: **Sample-level explanation.** We compare (a) DANCE with (b) existing attribution-based methods: GradCAM [46] attribution and Saliency Tubes [50].

Figure 5: **Visualization of model weights of similar action class pairs.** For each example, we show a Sankey diagram of the final layer weights associated with a similar action class pair.

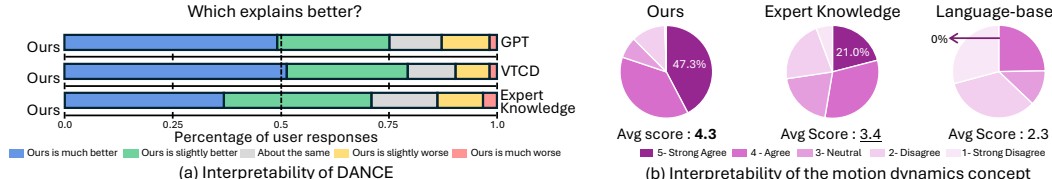

Figure 6: **Interpretability of DANCE.** (a) We present user study results from pairwise comparisons evaluating the interpretability of DANCE against three video XAI baselines: (i) a concept bottleneck model using spatio-temporal concepts generated by GPT-4o [19], (ii) VTCD [26], a spatio-temporal saliency-based explanation method, and (iii) a concept bottleneck model using spatio-temporal concepts from UCF-101 attributes [49]. (b) We report user study results comparing the interpretability of three concept types: (i) language-based concepts generated by GPT-4o, (ii) expert-defined concepts based on UCF-101 attributes [49], and (iii) our proposed motion dynamics concepts.

the action prediction $\hat{\mathbf{y}} = h(\mathbf{z}; \mathbf{W}_A)$, with a regularization term to enhance interpretability [58, 38]:

$$L_{\text{cls}} = -\frac{1}{K} \sum_{k=1}^{K} y_k \log(\hat{y}_k) + \lambda [(1-\alpha)\frac{1}{2} \|\mathbf{W}_A\|_F + \alpha \|\mathbf{W}_A\|_{1,1}], \tag{6}$$

where $\hat{y}_k$ denotes the $k$-th element of $\hat{\mathbf{y}}$, $\|\cdot\|_F$ represents the Frobenius norm, $\|\cdot\|_{1,1}$ is the element-wise $\ell_1$ norm, and $\lambda$ and $\alpha$ are balancing hyperparameters.

## 4 Experimental Results

In this section, carefully design and conduct rigorous experiments to answer the following research questions: (1) Does DANCE generate explanations that are easy for humans to interpret in the context of action prediction? (Section 4.1) (2) Can DANCE detect changes in the temporal domain, such as reversed input sequences? (Section 4.1) (3) What is the performance trade-off, if any, when interpretability is introduced into a previously non-interpretable model? (Section 4.2) (4) Can DANCE be effectively used for model *debugging* and *editing*? (Section 4.3) For more details on the dataset and implementation, please refer to the supplementary materials.

### 4.1 Analysis

In this section, we examine whether DANCE produces explanations that are easily interpretable for humans in the context of action prediction. To this end, we first define concept contribution to quantify the importance of each concept neuron, and then provide sample-level and model-level explanations, followed by the results of a user study evaluation. Additional visualizations and further details for this section are provided in the supplementary materials.

**Concept contribution.** We define the concept contribution as the product of a concept activation and the concept weight associated with the predicted class and we use this term consistently throughout the paper. To visualize a motion dynamics concept, we select the pose sequence closest to the cluster medoid and use it as the representative example.

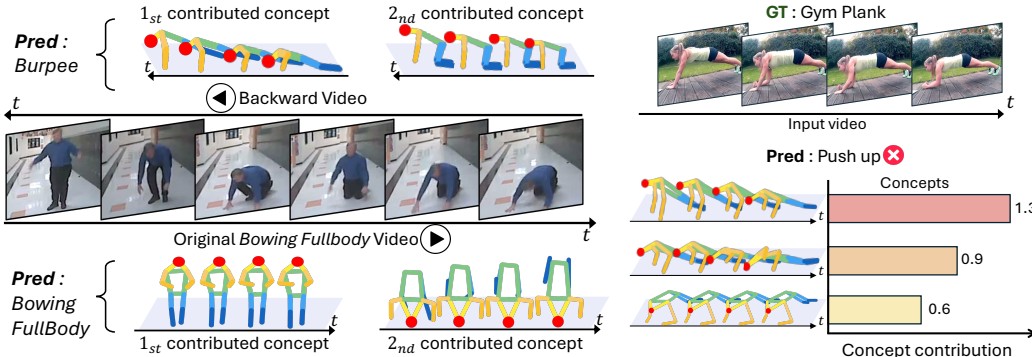

Figure 7: **Sanity check of DANCE.** We compare predictions and top-2 contributing concepts of DANCE for (i) the original video, and (ii) the same video played backward.

Figure 8: **Failure case analysis using DANCE.** With intuitive explanations, DANCE can help failure case analysis.

**Sample-level explanation.** In Figure 4, we visualize the top-3 contributing concepts along with the input video. In Figure 4 (a), DANCE leverages the motion concepts "lowering" and "lifting", along with the object concept "Bench", which together support the correct prediction. In contrast, as shown in Figure 4 (b), saliency-based methods [46, 50] produce spatio-temporally entangled explanations, making it unclear whether the model bases its decision on the object itself, i.e., "barbell" or its motion, i.e., up-down movement when predicting *Bench Press*.

**Model-level explanation.** In Figure 5, we visualize concept-to-class weights for pairs of similar action classes using Sankey diagrams. The thickness of each edge represents the relative contribution weight. DANCE provides the model's decision basis, helping users understand how it discriminates between similar actions. For example, in Figure 5 (a), *Basketball Shoot* and *Volleyball Set* share common motion concepts (shown in light khaki), while DANCE distinguishes between them using subtle motion differences (shown in pink) and the object concept, "Volleyball Poles."

**Interpretability of DANCE.** To evaluate the effectiveness of DANCE in explaining model predictions, we conduct a user study comparing it against three baseline explainable video action recognition methods: (i) a CBM using entangled spatio-temporal concepts generated by GPT-4o [19], (ii) VTCD [26], a spatio-temporal saliency-based method, and (iii) a CBM using spatio-temporal concepts from UCF-101 attributes [49]. In each question, we ask each participant a pairwise comparison question, by showing a video along with explanations from DANCE and one of the baselines: "Which explanation helps you better understand why the model predicted the action?" We collect responses on a five-point Likert scale, ranging from "DANCE is much better" to "the other method is much better." As shown in Figure 6 (a), more than 70% of the responses fall into "ours is much better" or "ours is slightly better" categories across all three pairwise comparisons. These results showcase that users perceive DANCE as more intuitive and trustworthy than (i) language-based explanations, (ii) unstructured saliency-based attributions, and (iii) expert-defined concepts. For more details, please refer to the supplementary materials.

**Interpretability of the proposed motion dynamics concept.** We conduct a user study evaluating the interpretability of three CBMs using the following temporal concepts: (i) language-based concepts generated by GPT-4o [19], (ii) expert-defined concepts based on UCF-101 attributes [49], and (iii) our proposed motion dynamics concepts. For each baseline, we randomly select one concept and visualize the top six video clips whose top-1 concept activation corresponds to the selected concept. We then presented the concept alongside the activated video clips, and asked participants the following question: "How well does the given concept match the actions or motions shown in the video?" We collect the response on a five-point Likert scale, where higher scores indicate stronger perceived alignment and interpretability. As shown in Figure 6 (b), our motion dynamics concept achieves the highest average score of 4.3, with 89.7% of participants rating it 4 or 5. In contrast, language-based and expert-defined concepts receive lower average scores of 2.3 and 3.4, respectively. These results indicate that the proposed motion dynamics concept is significantly more intuitive and aligned with human perception of motion, providing more interpretable explanations. For more details, please refer to the supplementary materials.

**Sanity check.** Here, we check the sanity of DANCE from the perspectives of model behavior and explanation quality. In Figure 7, we compare the predictions and top-2 contributing concepts of both methods for (i) the original video and (ii) the same video played backward. DANCE correctly predicts *Bowing FullBody* for the original video leveraging the visualized motion dynamics concepts. For the backward video, DANCE predicts *Burpee* by leveraging "standing up" like motions as visualized,

Table 1: **Video action recognition performance.** We report the Top-1 accuracy (%) of the baselines with and without interpretability as well as DANCE, all using the same backbone encoder [53].

| Method | KTH [45] | Penn Action [61] | HAA-100 [9] | UCF-101 [49] |
|---|---|---|---|---|
| Baseline w/o interpretability | 89.7 | 97.8 | 73.5 | 88.4 |
| CBM [25] w/ UCF-101 attributes | - | - | - | 86.8 |
| LF-CBM [38] w/ entangled language concepts | 87.4 | 96.3 | 66.5 | 85.5 |
| LF-CBM [38] w/ disentangled language concepts | 89.9 | 97.7 | 65.3 | 83.7 |
| DANCE | 91.1 | 98.1 | 70.7 | 87.5 |

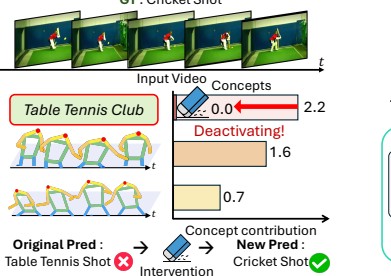

Figure 9: **Sample-level intervention.** Deactivating the irrelevant concept leads to fixing the misprediction to a correct predcition.

Figure 10: **Cross domain class-level intervention.** Fixing zero weights from relevant concepts to classes result in 4.3 point accuracy improvement under *severe domain shift*, i.e., UCF-101 [49] → UCF-101-SCUBA [29], without retraining.

reflecting the model's sensitivity to temporal direction. With the sanity check, we validate that (i) the model is sensitive to motion dynamics as intended, and (ii) explanations by DANCE are interpretable.

**Failure case analysis.** Thanks to the transparent and intuitive nature of DANCE, we can also *analyze* the reason behind incorrect predictions as shown in Figure 8. DANCE explains that the misprediction of *Push up* is because of high contributions from downward body movement. Such interpretability enables targeted model debugging, as further demonstrated in Section 4.3.

## 4.2 Performance Evaluation

**Interpretability and performance do not always trade-off.** We compare DANCE with a baseline model without interpretability, as shown in Table 1. DANCE achieves slightly higher accuracy on KTH and Penn Action, while showing a modest drop of 2.8 points on HAA-100 and 0.9 points on UCF-101. These results indicate that DANCE can deliver intuitive and structured explanations without substantially compromising classification performance—and in some cases, even improving it.

**Clearer concepts improves performance.** We further investigate whether the clarity of concept representations affects model performance. To this end, we compare DANCE with the following baselines: (i) concept bottleneck model with UCF-101 attributes [49], (ii) a label-free concept bottleneck model [38] using spatio-temporally entangled concepts generated by GPT-4o, and (iii) a variant of the label-free concept bottleneck model that uses the same object and scene concepts as DANCE, but uses GPT-4o-generated temporal concepts. Across all datasets, DANCE consistently outperforms these baselines, demonstrating that employing clearer and disentangled concept representations—particularly for motion dynamics—can lead to improved action recognition performance.

## 4.3 Model Editing

Here, we demonstrate the utility of DANCE in *debugging* itself by analyzing sample-level concept contributions in misclassifications and inspecting class-level weights. We provide additional results of model editing in the supplementary materials.

**Sample-level intervention.** In Figure 9, we illustrate how a user can intervene the model by removing a specific concept in the case of misclassification. For example, in Figure 9 (a), the model initially predicts *Table Tennis Shot*, largely influenced by the scene-level concept "Table tennis club." When this concept is deactivated, the model leverages motion dynamics concepts and correctly classifies the input as *Cricket Shot*. The results demonstrates that DANCE supports fine-grained, transparent control over predictions, allowing users to actively adjust model behavior.

**Cross domain class-level intervention.** In Figure 10, we demonstrate class-level intervention using DANCE. This experiment evaluates whether such adjustments can resolve performance drop caused by *severe distribution shift*. We evaluate on UCF-101-SCUBA [29], a variant of UCF-101 [49] where test video backgrounds are altered to induce a domain shift. As shown in Figure 10, for the

*Volleyball Spiking* class, the model activates a relevant motion dynamics concept but ignores it due to a zero weight. By assigning a weight of 1.0 to this concept, we correct 98 misclassifications with only one additional error, significantly improving overall accuracy by 2.5 points ($84.0\% \rightarrow 86.5\%$). Further adjusting weights for the *Golf Swing* and *Tennis Swing* classes result in a 4.3 point accuracy improvement ($77.7\% \rightarrow 82.0\%$). These findings highlight DANCE's ability to support post hoc model *debugging* and performance recovery under severe domain shifts, *without retraining*.

## 5 Conclusions

In this paper, we propose DANCE to address the challenge of explaining video action recognition models in a structured and motion-aware manner. DANCE grounds its predictions in three human-interpretable concept types—motion dynamics, objects, and scenes—enabling cognitively aligned and transparent explanations. This design facilitates intuitive understanding of model behavior by explicitly separating temporal and spatial reasoning. Through extensive experiments and practical use cases, we show that DANCE delivers clearer and more faithful explanations while maintaining competitive recognition performance. Moreover, DANCE supports effective model editing—even under severe domain shifts—without requiring retraining. We believe our work offers valuable insights to the XAI and video understanding communities and will help inspire future research.

**Acknowledgment.** This work was supported in part by the Institute of Information & Communications Technology Planning & Evaluation (IITP) grant funded by the Korea Government (MSIT) under grant RS-2024-00353131 (20%), RS-2021-0-02068 (Artificial Intelligence Innovation Hub, 20%), and RS-2022-00155911 (Artificial Intelligence Convergence Innovation Human Resources Development (Kyung Hee University, 20%)), Additionally, it was supported by the National Research Foundation of Korea(NRF) grant funded by the Korea government(MSIT)(RS-2025-02216217, 20%) and (RS-2025-22362968, 20%). We thank Gangmin Choi, Kyungho Bae and Yong Hyun Ahn for the valuable discussions and feedback.

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
