# OpenReview forum: "Disentangled Concepts Speak Louder Than Words: Explainable Video Action Recognition"
_NeurIPS.cc/2025/Conference — NeurIPS 2025 spotlight_

### Official Review · Reviewer_g8Sg · 2025-06-24

**Clarity:** 3
**Significance:** 3
**Originality:** 3
**Rating:** 5
**Confidence:** 3

**Summary:**

The authors propose the Disentangled Action aNd Context concept-based Explainable (DANCE) framework.
The goal is to provide concept-based explanations, which can belong to one of three categories: motion dynamics, objects, or scenes. Motion dynamics concepts are automatically discovered by clustering pose sequences. They assess the effectiveness of their method through experiments conducted on four distinct datasets.

**Questions:**

1. Do you have results for DANCE model performance when excluding a specific type of concep, such as the motion dynamics category, to demonstrate its necessity?

2. You mention a baseline model (lines 295–296), but I am not entirely sure what this model entails or how it differs from the model proposed in the DANCE framework. Could you please clarify?

3. What was the motivation for adopting a two-stage training process, as described on lines 210–211?

**Ethical Concerns:**

["NO or VERY MINOR ethics concerns only"]

**Final Justification:**

The authors have addressed all of my questions, and as a result, I have increased my rating.

**Limitations:**

Yes

**Quality:**

3

**Strengths And Weaknesses:**

**Strengths :**
The method section is clear, with a key contribution being the introduction of the motion dynamic concept category and demonstrating how these concepts can be discovered within the model without relying on human annotation. Additionally, a notable strength is the inclusion of a user study to evaluate the quality of the explanations in comparison to other methods.

**Weaknesses:** It seems that the main contribution lies in the disentangled explanations and the introduction of the motion dynamics concept category. Therefore, I would have expected to see results demonstrating the benefit of incorporating these additional concepts (see Question 1).
In Table 1, I am not sure I fully understand what the baseline model represents (see Question 2).

---

> ### Author Rebuttal · Authors · 2025-07-29
>
> ### **Contribution of Disentangled Concept Types**
>
> We thank the reviewer for the insightful question regarding the contribution of each concept type. We agree that analyzing the effect of each concept type is important, and we have already conducted such an ablation study on the UCF-101 dataset. We reported in Table 9 of the supplementary material. For clarity, we summarize the key findings below.
>
> Using only **motion dynamics concepts** achieves **86.4%** accuracy, demonstrating their strong standalone importance. **Adding object or scene concepts** on top of motion dynamics increases accuracy to 86.7% and 86.5%, respectively—indicating that contextual information provides complementary cues, with object concepts offering slightly more action-relevant signals than scene context. In contrast, using **only object and scene concepts** without motion dynamics results in a notable performance drop to **84.6%**, highlighting the limitations of relying solely on spatial context for recognizing video actions.
> Finally, combining all three concept types—motion dynamics, object, and scene—achieves the highest accuracy of **87.5%**, confirming the complementary nature of **disentangled spatio-temporal concepts**.
>
> These results validate the importance of motion dynamics and demonstrate the effectiveness of our structured concept representation in improving both interpretability and classification performance.
>
> ### **Clarification on the Baseline Model in Table 1**
>
> We apologize for the lack of clarity regarding the baseline model reported in Table 1 of the main paper. The entry *Baseline w/o interpretability* refers to a standard video recognition model that does not incorporate any concept supervision. Specifically, we fine-tune a video encoder  (VideoMAE) and a classification head directly on each target dataset **without any concept supervision**. For fair comparison, we use Something-Something-v2-pretrained VideoMAE weights for KTH, Penn Action, and HAA-100, and UCF101-pretrained weights for UCF-101, using the same pretrained weights as DANCE. We note that the VideoMAE used in *Baseline w/o interpretability* is identical to the video feature extractor employed in DANCE, ensuring a fair backbone comparison.
>
>
> The key difference from DANCE lies in **the absence of the concept layer**, which in our framework consists of three disentangled types: motion dynamics, object, and scene concepts. This baseline follows a common protocol in recent CBM-based approaches[1-4], where the backbone remains the same and only the concept layer is removed.
>
> Therefore, this non-interpretable setup serves as a reference point to assess **the trade-off between accuracy and interpretability** introduced by DANCE. We will clarify this setup explicitly in the final version of the paper.
>
> ### **Motivation for adopting a two-stage training process**
>
> Thank you for your question regarding our two-stage training strategy. As described in lines 210–211 of the main paper, we first train the concept prediction layers, and then train the linear classification layer while keeping all concept layers frozen. In general, there are **two possible training paradigms** in concept bottleneck models:
>
> (i) *sequential (two-stage)* — first train the concept layer, then the classification layer (our approach),
>
> (ii) *joint* — train both simultaneously using a total loss ( e.g., $L=L_{concept}+L_{classification}$ ).
>
> We adopt the **two-stage** approach primarily due to our emphasis on explainability over pure classification performance. In joint training, the classification loss can influence the concept layer, potentially introducing shortcuts that prioritize end-task performance at the cost of faithful concept learning. By separating the training of the concept and classification layers, we encourage the concept layers to capture semantically **meaningful representations independent of the final labels**.
> This motivation aligns with prior findings in the original CBM paper [1], which shows that sequential training better focuses on concept learning compared to joint training. Moreover, many recent CBM-based approaches adopt two-stage training for similar reasons [1-4].
>
>
>
>
> #### **References**
> [1] Koh, Pang Wei, et al. "Concept bottleneck models." ICML 2020.
>
> [2] Oikarinen, Tuomas, et al. "Label-free Concept Bottleneck Models." ICLR 2023.
>
> [3] Yuksekgonul, Mert, Maggie Wang, and James Zou. "Post-hoc Concept Bottleneck Models." ICLR 2023.
>
> [4] Shang, Chenming, et al. "Incremental residual concept bottleneck models." CVPR 2024.

---

### Official Review · Reviewer_JS41 · 2025-07-02

**Clarity:** 4
**Significance:** 3
**Originality:** 2
**Rating:** 4
**Confidence:** 4

**Summary:**

The paper proposes DANCE, an ante-hoc, concept-bottleneck framework for video action recognition that forces the model to reason through three disentangled and human-interpretable concept spaces: motion dynamics, objects, and scenes. Motion concepts are created without labels by clustering 2-D pose sequences extracted from automatically detected key clips, giving appearance-invariant representations of how the body moves. Object and scene concepts are obtained by querying GPT-4o for class-specific keywords and then pseudo-labeled in videos via a vision–language dual encoder.

**Questions:**

- How do you use GPT-4o for videos, I thought it does not allow videos. Do you pass in key image frames and the model handles it as separate images?
- LLM-generated lists are ad-hoc, prompt-dependent, and not guaranteed to be exhaustive. Rare but crucial objects may be absent. How do you handle if your test videos has something unseen yet?
- Forcing predictions through a relatively small concept basis would hurt recognition, for instance if a new shape appears, the model must either conflate it with an existing object concept or ignore it. Does accuracy degrade on rare subclasses?
- For the CLIP thresholding you mention, how is thresholding of CLIP similarity performed to binarise object/scene labels? Is it class-agnostic or tuned per concept?
- How many pose sequences and how many FINCH clusters per dataset? It would be very useful to analyze this since we probably do not want a lot of clusters.

**Ethical Concerns:**

["NO or VERY MINOR ethics concerns only"]

**Final Justification:**

I think quite a few of my questions are addressed, some limitations like what clusters are used and the method only working for 1 human are still there like i indicated earlier. i had anyways given a high score and so i stick to it.

**Limitations:**

- Gains of +1% on small datasets (like shown) can easily fall within variance from random initialisation or sampling. No standard deviation across seeds is reported.
- One big limitations is all the clusters are precomputed and it is very hard to generalize outside what you have already seend uring making the clusters.
- How does DANCE handle actions that involve two coordinated agents (e.g., handshake) or tool–object interaction where the object itself moves (e.g., “pull-ups on bar”)? The current motion representation is strictly single human-centric.
- Pose snippets are fixed-length (L frames). Actions whose discriminative cue lasts longer than L are split across several clusters and the bottleneck has no mechanism to model their ordering.
- Skeleton co-ordinates are pixel-based, not camera-normalised. Without 2-D/3-D normalisation, clusters entangle action with camera geometry.
- Motion clusters and GPT lists are dataset‐specific.Porting the model to a new domain (e.g., underwater sports apart from just sports like the examples) requires re-running both pipelines and retraining all heads which make it hard to follow the argument that “concepts give modularity”. The framework also does not work for anything different than one human doing some task in a exocentric view.

**Paper Formatting Concerns:**

I did not notice any.

**Quality:**

3

**Strengths And Weaknesses:**

## Strengths

- The paper tackles the under-explored but important question of how to produce human–aligned explanations for video action recognition, explicitly separating the “how” (motion) from the “what/where” (object- and scene-level context).
- By making the solution as an ante-hoc concept bottleneck with three disjoint partitions, the authors provide a simple yet expressive inductive bias that is easy to reproduce and to reason about.
- Object and scene concepts are sourced automatically with a single prompt to an LLM, eliminating any handrcrafting which you often see in many concept-based models.
- The pipeline is modular: each component (pose estimator, clustering algorithm, vision–language encoder) can be swapped for upgrades without retraining the full network.

## Weaknesses

- The novelty is limited. The architecture is a textbook concept-bottleneck pipeline (linear projection -> sparse linear head) is quite standard. Also disentangling the bottleneck into three blocks does not guarantee disentanglement in the learned representation. The main changes the authors make are mainly highly handcrafted ways (this causes man of the other limitations and weaknesses) of disentangling into 3 components.
- Two-stage training (first concepts, then classifier) does not preclude “shortcut” interactions between concepts at evaluation time. A stronger test would be to interchange concept vectors between unrelated videos and show that predictions change in an intuitive way.
- Clustering 2-D poses from short “key clips” is heuristic. Clusters might mix mutually incompatible motion primitives (e.g., mirrored left/right variants) or fragment a single semantic motion across multiple clusters. Something like mutual information with ground-truth actions, intra-cluster variance, or human-label agreement is needed to verify that the clusters are indeed coherent though the results shown actually look good.
- Similar to playing a clip backwards which is a decent diagnostic, but the paper does not try more extrem tests like re-colouring frames, jittering the bounding box, or removing the person entirely would test whether the model truly disentangles motion from context.
- FINCH requires pairwise nearest-neighbour computations over all pose sequences. For Something-Something or Kinetics this will be really large, is that why you only tests on smaller datasets?
- Nothing prevents the model from silently exploiting residual information in the frozen backbone that never surfaces in the linear concept layer. SImilarly, when parent concepts are correlated (e.g., “baseball-bat” always co-occurs with “swing-pose”), gradients can flow through a single node and the classifier still reaches high accuracy.
- The linear concept layer is extremely overcompletefd (hundreds of motion clusters + hundreds of GPT objects/scenes) but the only sparsity pressure is a l1 term on the classifier, not on the concept activations themselves. Activations therefore remain dense, defeating the “bottleneck” intuition and hampering faithfulness.

---

> ### Author Rebuttal · Authors · 2025-07-30
>
> ### **Clustering**
> #### **Key clips**: We determine $L$ for each dataset based on the overall video lengths and validation performance, as detailed in Supp. Table 4. To better capture discriminative cues in longer actions, we plan to incorporate temporal action localization methods that propose informative segments. Given a proposal, we can uniformly sample $L$ frames within the localized region, ensuring that key motion information is preserved without increasing model complexity.
>
> #### **Fragmented motions**: Regarding fragmented motions across multiple clusters, we clarify that this is a deliberate design. Actions like a "long jump" involve distinct motion phases (e.g., running and jumping), which we capture as separate motion dynamics concepts. This modularity supports compositional generalization and encourages concept reuse across different actions, e.g., reusing a running concept for "high jump".
>
> #### **Mirrored motions**: Through manual inspection, we find some clusters contain mirrored variants (e.g., left- and right-handed motions). To address this, we increase the number of pose clusters from 0.5K to 2K on UCF-101, and we observe that left/right-specific clusters naturally emerged—suggesting that clustering granularity can help capture such variations when needed.
>
> #### **Coherence**: On Penn Action, we cluster 14K pose sequences into 80 groups using FINCH and evaluate concept coherence via NMI with action labels—0.62—indicating strong semantic alignment. Intra-cluster variance (12.1±5.5) is much lower than global variance (66.2), showing tight grouping. A human study with 50 participants yields a 4.2 ± 0.7 Likert score on visual similarity, confirming high perceived coherence.
>
> #### **Scalability**: FINCH takes 49.3 seconds on UCF-101 (10K sequences) and 200.9 seconds on Kinetics-400 (240K sequences), which remains efficient and does not significantly impact the overall pipeline. This is because the original FINCH paper provides an approximate nearest neighbor (ANN) method, reducing the complexity from $O(N^2)$ to $O(N log N)$. While we can run FINCH clustering on Kinetics-400 or Something-Something-V2, we do not run our model on these datasets because of data-quality considerations, e.g., humans not visible, unreliable pose estimation due to occlusion. We plan to extend our work to more general scenarios in future work.
>
> #### **Precomputed clusters**: Our motion dynamics concepts capture reusable primitives across classes—for example, “down-swing” appears in both *badminton overswing* and *volleyball overhand* (Supp. Fig. 13). To assess generalization, we simulate a class addition scenario on UCF-101 by holding out one class during training. We then assign its pose sequences to existing clusters without retraining. DANCE achieves 87.1% accuracy after adding the class, compared to 87.2% before—a negligible drop. This suggests that the clusters generalize well to unseen classes.
>
> ### **Shortcut**
> To directly test for shortcut interactions, we perform concept interchange experiments on Penn Action.
>
> **Scenario I: Similar contexts, different motions**: We swap motion dynamics between *Tennis forehand* and *Tennis serve*, which share scene/object context. The predicted classes also swap with high confidence, demonstrating that DANCE bases its decision on motion concepts rather than static context, confirming their disentangled influence.
>
> ||**Motion dynamics swap in a *tennis forehand* video**||
> |-|:-:|:-:|
> || Tennis forehand conf.| Tennis serve conf.|
> |Before| 0.98|0.01|
> |After|0.01|0.98|
>
> || **Motion dynamics swap in a *tennis serve* video**||
> |-|:-:|:-:|
> || Tennis forehand conf.| Tennis serve conf.|
> |Before| 0.01|0.99|
> |After|0.98|0.01|
>
> **Scenario II: Unrelated actions**: We swap scene concepts between *Golf swing* and *Bench press* videos. Golf swing’s confidence drops notably due to its unique spatial context within the Penn Action dataset, while Bench press is unaffected as it shares context with other gym actions where motion dominates (e.g., Squat and Clean & Jerk), within the dataset. This intuitive response to concept swaps shows that DANCE does not exploit shortcuts and that its predictions are grounded in the appropriate concept type.
>
> ||**Scene swap in a *golf swing* video**||
> |--|:-:|:-:|
> || Golf swing conf.  | Bench press conf.|
> |Before| 0.98|0.01|
> |After|0.54|0.01|
>
> || **Scene swap in a *bench press* video**||
> |--|:-:|:-:|
> || Golf swing conf.  | Bench press conf.|
> |Before| 0.00|0.99|
> |After|0.00|0.99|
>
> ### **Concept disentanglement**
> In Supp. Section 6.2 and Figure 16, we create static videos by replicating a single frame across time. For example, replicating a handstand frame from a balance beam video leads DANCE to predict HandstandWalking—a class with a similar pose but with a different context. The result suggests that DANCE successfully disentangles motion from spatial context and makes predictions based on the underlying motion dynamics.
>
> In addition, we apply color jittering to every frame of a normal video during inference on UCF-101. Specifically, we randomly perturb brightness (±40%), contrast (±40%), saturation (±40%), and hue (±20%) for each frame. The overall top-1 accuracy slightly drops from 87.5% to 86.3%, indicating that DANCE remains robust even under significant appearance distortions.
>
> ### **Residual information**
> We mitigate leakage by freezing the backbone and enforcing a strict bottleneck: predictions are made only from a sparse linear layer. While some concept correlations (e.g., “baseball-bat” and “swing-pose”) may persist, our framework enables *explicit inspection* and *intervention*, improving transparency despite imperfect disentanglement.
>
> As shown in Fig. 10 (UCF101-SCUBA), both VideoMAE and DANCE initially perform poorly (<4%) due to context bias. However, manual edits to DANCE recover 72 samples, raising accuracy to 35.0%, while VideoMAE remains unchanged. This demonstrates DANCE’s practical advantage: user-guided correction enabled by transparency, even when a model inherits bias from the frozen encoder.
>
> ### **Sparsity**
> We observe that classifier sparsity remains below 5% across all experiments, indicating reliance on a small set of meaningful concepts. At the sample-level explanation (e.g., Fig. 11, “punching sandbag” Supp.), the top-5 concepts contribute 9.95 to the logit, while the remaining 495 contribute only 0.8—under 8%. This confirms that predictions are driven by a few essential concepts, supporting faithfulness.
>
> ### **Novelty**
> While our framework adopts the standard CBM architecture, our key novelty lies in extending CBMs to the *video domain* through a *structured, interpretable decomposition* into motion dynamics, object, and scene concepts. This tripartite disentanglement is non-trivial in spatio-temporal settings and well aligned with human cognition. Furthermore, our approach uses unsupervised motion concept discovery and zero-shot LLM supervision for object and scene concepts, which together enable interpretable representations without manual annotation or task-specific and heavy engineering.
>
> ### **GPT-4o for videos**
> We input class names as text prompts, not images or videos, to generate related object and scene concepts. See Section 1.3.2 and 4.2 of Supp. for more details.
>
> ### **Rare objects**
> While LLM-generated concepts may miss rare objects, GPT-4o yields robust and representative concepts from action names alone, outperforming LLaVA, which uses key frames for concept extraction (84.6% vs. 82.9%; see Supp. Table 5, Sec. 6.1.1).
> Moreover, DANCE’s modular design allows it to fall back on available concept types. As shown in Fig. 10, it remains effective under domain shifts by relying solely on motion dynamics. Even when all concepts fail, DANCE still supports interpretable failure analysis—unlike black-box models.
>
> ### **Small concept basis**
> We acknowledge that a small concept basis may impact recognition. However, as shown in Supp. Table 6, increasing motion concepts yields only marginal gains, suggesting sufficiency. To test robustness on rare subclasses, we group UCF-101 into head/mid/tail classes. DANCE achieves 87.5% overall, with 87.9%, 85.2%, and 89.5% on head, mid, and tail groups—demonstrating strong performance even on underrepresented classes.
>
> ### **CLIP thresholding**
> We do not binarize object or scene concept labels but treat them as soft labels ranging from 0 to 1. Instead, we apply fixed threshold-based filtering at two stages: (1) initial activation filtering that removes concepts with low average ViCLIP activations and (2) projection filtering that removes low projected concept activations after optimization. This class-agnostic strategy follows LF-CBM and involves no per-concept or per-class tuning. See Supp. Sections 1.3.2 and 2.2 for details.
>
> ### **Pose sequences**
> FINCH yields a hierarchy of motion concepts at varying granularities. For UCF-101, ~100K pose sequences produce six levels, from 22,706 fine-grained to 10 coarse clusters. This allows flexible abstraction levels when selecting motion concepts.
>
> ### **STD**
> We report the mean and standard deviation across 5 random seeds as follows.
>
> |KTH|Penn Action|HAA-100|UCF-101|
> |:-:|-:|-:|-:|
> |91.18±0.02|98.08±0.00|70.76±0.65|87.58±0.10|
>
> ### **Skeleton**
> As described in Section 1.3.1 Supp., we normalize each pose by centering the joint coordinates and scaling them to the range [0, 1].
>
> ### **Two agents**
> We focus on single-human actions as a first step toward explainable video recognition. While our current framework does not yet handle multi-agent or object-centric motions, future work will explore multi-person inputs, relation-aware clustering, and segment-based trajectory encoding (e.g., SAM2 [Ravi et al, ICLR 2025] ) to generalize motion dynamics beyond single-human scenarios.
>
> ### **New class/domain**
> Please refer to the fourth paragraph of our response to Reviewer nxgA: Flexibility-interpretability trade off.

---

> > ### Comment · Reviewer_JS41 · 2025-08-07
> >
> > thanks for the responses.
> >
> > i was anyways recommending acceptance and i still recommend acceptance.

---

### Official Review · Reviewer_DRfP · 2025-07-03

**Clarity:** 3
**Significance:** 3
**Originality:** 3
**Rating:** 5
**Confidence:** 3

**Summary:**

DANCE, the method, inputs a video and aims to explainably classify the action occuring in said video. Each sample video during training also includes a binary presence/absence vector for concepts.

A pre-trained encoder is used to extract features from the video. These features are projected with a linear concept layer into M activations (where M is the number of concepts), with each projection corresponding to a concept. This concept projection layer has weights that correspond to different types of concepts, specifically motion dynamics, objects, and scene concepts. The goal of this process is an explainable "concept activation" vector of size M. This vector is apportioned into a section related to motion dynamics, objects, and scenes. This vector is then used to predict a final action.

Prior to training this method, concept discovery is done. To get motion related concepts, the authors use key clip selection to find chunks with motion cues. They estimate 2D pose and for each frame in a clip of length L and then cluster the motion patterns. After doing this across the training dataset, they can extract clusters as motion concepts. For the concept activation vector with activations related to motion, each index relates to these clustered motion concepts. For each training video, that video/clip has that motion concept annotated if it belongs to that cluster.

Beyond motion concepts, object and scene concepts are generated for action classes by using VLMs. Specifically, an LLM first pseudo-annotates what objects are related to an action class, and then vision-language dual encoder is used to determine if those objects are present in the clip. Similarly to motion concepts, the clip is then annotated with the presence/absence of these objects through a similarity score between features.

Overall, the paper finds ways to pseudo-annotate motion, object, and scene concepts in video clips. These pseudo-annotations come from discovered concepts in both the data and how the data relates to an LLM's understanding of what the actions should involve. After concept discovery, the presence of these concepts is represented as part of the sample.

Next, training is done to predict the presence/absence of concepts independently of the action prediction. After this training is complete, the concept layer is frozen and the final classifier is trained. This separation of concerns leads to an interesting and interpretable way of understanding how concept activations lead to action classifications, as the concept scores are introspectable.

**Questions:**

LLMs / VLMs were used for data annotation, couldn't they have been used for an additional baseline? It would be nice to see performance of more modern / more broad methods.

Although language is a poor description of motion as shown in Figure 1, I'm curious how a method that was solely based on language would perform. Other prior work on relationship detection and other related action topics have shown strong priors based on the presence/absence of objects and/or just simply using language models on textual descriptions of the videos.

**Ethical Concerns:**

["NO or VERY MINOR ethics concerns only"]

**Final Justification:**

The authors adroitly responded to my review. The paper is interesting and the other reviewers agreed on our ratings.

**Limitations:**

Motion clustering has implications for surveillance as it provides an easy data-centric way to extract concerning human poses. This type of easy-to-detect pose sequence clustering could have negative impacts as it is a lot easier to run this clustering on the outputs of a 2D pose detector over time than it is to process all the video in other more computationally expensive ways.

**Quality:**

3

**Strengths And Weaknesses:**

This paper follows a tried and true methodology of data-centric concept discovery leveraged in the training of a classifier for use in a downstream task. The ability to pseudo-label annotate the concepts of the training data in order to first train a concept predictor and then action predictor is useful for creating an interpretable model.

The user study shows good performance for this method, and it is also useful quantitatively as it performs on par with the baseline w/o interpretability.

Rich interpretability as proposed in DANCE is useful both for better introspection into why/how classifiers make decisions and when a data-centric concept-learning phase as used in DANCE can actually help outperform uninterpretable baselines by grounding the focus of the classifier on certain relevant/salient distinctions (such as motion categories). Overall, this paper is quite interesting and helps expand the XAI domain.

Weaknesses include the limitation of the motion-related concepts to human pose. Ideally, motion concepts could be more general and not just human-centric (although this is being used for action recognition).

---

> ### Author Rebuttal · Authors · 2025-07-30
>
> ### **Toward generalization beyond human-centric motion dynamics concept**
>
> We sincerely thank the reviewer for highlighting the limitation regarding the scope of our motion dynamics concepts. We understand and appreciate the concern that relying solely on human pose may restrict the generality of motion representation.
> In this work, we focus on explaining *human actions* in videos, where pose sequences provide structured and semantically meaningful cues for motion understanding.
>
> Given our focus, we adopt human pose sequences to capture *fine-grained temporal patterns*. This representation enables users to intuitively understand how an action unfolds over time without being distracted by irrelevant visual factors such as clothing or background. Importantly, we position this work as an initial step toward ante-hoc explainable video understanding. Our goal is to establish a clean and modular framework grounded in disentangled concept types—motion dynamics, object, and scene—that allows structured, faithful, and intuitive explanation. Starting with human pose is a practical and task-aligned choice that provides a solid foundation for further development.
>
> #### **Limitations in non-human-centric videos**
> In scenarios where humans are not present, DANCE can still produce explanations based on object and scene concepts. While these allow for reasonable interpretation of the spatial context, they may be less effective at capturing fine-grained object motions or physical interactions. We acknowledge this limitation and further discuss it in the limitations and broader impact section of the supplementary materials.
>
> #### **Future directions for generalizing motion dynamics concept**
> Therefore, we agree with the reviewer that extending motion dynamics concepts beyond human pose is an important and exciting direction. We plan to explore non-human-centric motion representations such as flow-based dynamics and object trajectories.
> Concretely, one possible approach is to leverage a video segmentation model such as Sam2 [1] *to capture the movement of salient objects* over time and define such patterns as motion concepts. These non-human-centric motion dynamics can be incorporated alongside our motion dynamics concept to create a more structured and general representation of motion. This unified formulation would allow DANCE to reason about both human actions and object-level dynamics in a coherent and interpretable manner.
> We hope that this work encourages further exploration in the underexplored area of ante-hoc explainable video models.
>
> ### **Additional baseline**
>
> We appreciate the reviewer’s suggestion to consider more modern and broad models such as vision-language models (VLMs) as baselines. To address this, we provide a comparison with InternVideo2 [2], a large-scale VLM trained on extensive video-text pairs, which supports both zero-shot inference via text generation and classification via linear-probing.
>
> | Method|Interpretability |Top-1 Accuracy|
> |--|:-:|:-:|
> | InternVideo2 zeroshot  |× |89.5|
> | InternVideo2 visual encoder|× |97.3|
> |  DANCE with InternVideo2 visual encoder |✓|  96.5|
>
> As shown above, InternVideo2 achieves strong performance (89.5% zero-shot ). While such models can produce textual rationales via prompting (e.g., chain-of-thought), these outputs are not faithful to the model’s internal reasoning and are hard to validate [3-5]. Additionally, current VLMs still lack fine-grained temporal understanding, limiting their suitability for structured, explanation-driven action recognition (see the last two paragraphs of Section 6.2 in the supplementary).
> Therefore, rather than using VLMs as standalone explainable models, we take a more practical and structured approach by employing their strong visual encoders within the DANCE framework, which is inherently interpretable. Specifically, we employ  InternVideo2’s visual encoder as the feature extractor $f(\cdot)$ in DANCE while keeping the rest of the pipeline unchanged. This variant achieves 96.5% accuracy—comparable to the original InternVideo2—while offering concept-based interpretability. This result demonstrates the scalability of our framework and its ability to integrate strong backbones like InternVideo2 without sacrificing explainability.
>
> ### **Comparison with "language-based explanation baselines"**
>
> In this work, we evaluate language-based explanation baselines, as shown in Figure 1 (b), Table 1, Figure 6, and the user study section (L258-282) of the main paper. Specifically, the method in Figure 1 (b) follows the Label-Free CBM (LF-CBM) [9] pipeline, where spatial and temporal concepts are derived directly via prompt engineering from GPT-4o, without manual annotations.
>
> To assess their effectiveness, we conduct a user study comparing explanations from LF-CBM with those of our motion dynamics concepts (Figure 6 of the main paper). For a fair comparison, both methods use the same video backbone (VideoMAE), ensuring that differences arise solely from the concept extraction process. In summary, our user study shows that motion dynamics concepts offer more intuitive and structured explanations than language-based ones, achieving higher absolute ratings (**Ours (4.3) vs Language-based (2.3) in Figure 6 (b)**) and **75% preference** relative to "GPT" (language-based baseline) in Figure 6 (a).
>
> Implementation details and evaluation protocols are provided in Sections 4.2 and 5.1 of the supplementary material. Additional qualitative examples of language concept-based explanations and discussions are in Figure 9, Figure 10, and L616-646 of the supplementary material. We hope this clarifies how language-only concept supervision is incorporated and evaluated in our framework.
>
>
> ### **Language-based and relationship detection approaches**
>
> While these methods show promising performance by aligning visual content with textual semantics, DANCE differs from them in key aspects.
>
> **(1) Relationship detection methods** typically require fine-grained relational labels (e.g., subject–object–action triplets) and rely on vision-language grounding to detect human-object interactions (HOIs) [6–8] . However, such annotations are not available in general action recognition datasets like UCF-101 or Penn Action, limiting their scalability without additional manual labeling. Moreover, most of these methods do not explicitly aim to provide interpretability.
>
> **(2) Language-only video descriptions** (e.g., using captioning models or VLMs) can generate high-level textual summaries, but they still operate in a black-box manner and do not provide faithful explanations of why a specific prediction was made [3-5]. Their descriptions serve as task outputs—summarizing content or enabling retrieval—rather than providing transparent access to the model’s internal decision-making process. As such, they belong to a different category of research, focused more on generation or alignment than explanation. In contrast, our work focuses on making the model’s reasoning process explicit and structured through disentangled concept representations, aligning more closely with the goal of faithful and ante-hoc interpretability.
>
> ### **Goal of our work**
> Rather than comparing DANCE explanations with the final outputs of language-based models, we compare them with explanations from language-based Concept Bottleneck Models (CBMs), which aligns better with our goal of making the model’s internal reasoning transparent.
>
> While recent works [9–11] in the image domain explore CBMs using language-derived concepts (e.g., CLIP embeddings), we argue that such representations are limited in the video domain due to the lack of temporal modeling and implicit motion cues. To address this, we propose motion dynamics concepts grounded in pose-based temporal patterns, complementing object and scene concepts from vision-language models. This structured, disentangled design enables DANCE to produce faithful, motion-aware explanations tailored for video understanding.
>
> ### **Concern about surveillance**
> We acknowledge the potential for misuse. However, our method builds on standard pose estimation and unsupervised clustering—tools already widely available—and does not infer intent or sensitive attributes. Its design centers on interpretability, not surveillance. We will include a disclaimer on responsible use in the broader impact section.
>
>
>
> #### **References**
> [1] Ravi, Nikhila, et al. "SAM 2: Segment Anything in Images and Videos." ICLR, 2025.
>
> [2] Wang, Yi, et al. "Internvideo2: Scaling foundation models for multimodal video understanding." ECCV 2024
>
> [3] Wang, Boshi, et al. "Towards Understanding Chain-of-Thought Prompting: An Empirical Study of What Matters." ACL 2023.
>
> [4] Arcuschin, Iván, et al. "Chain-of-Thought Reasoning in the Wild is not Always Faithful." Reasoning and Planning for LLMs Workshop, ICLR 2025.
>
> [5] Turpin, Miles, et al. "Language models don't always say what they think: Unfaithful explanations in chain-of-thought prompting." NeurIPS 2023.
>
> [6] Wang, Ning, et al. "Language model guided interpretable video action reasoning." CVPR 2024.
>
> [7] Zhao, Long, et al. "Unified visual relationship detection with vision and language models." ICCV 2023.
>
> [8] Wang, Yongqi, et al. "End-to-end open-vocabulary video visual relationship detection using multi-modal prompting." TPAMI 2025.
>
> [9] Oikarinen, Tuomas, et al. "Label-free Concept Bottleneck Models." ICLR 2023.
>
> [10] Yuksekgonul, Mert, Maggie Wang, and James Zou. "Post-hoc Concept Bottleneck Models." ICLR 2023.
>
> [11] Shang, Chenming, et al. "Incremental residual concept bottleneck models." CVPR 2024.

---

> > ### Comment · Reviewer_DRfP · 2025-08-06
> >
> > Thank you for addressing a number of my questions in your rebuttal! The explanations are clear.

---

### Official Review · Reviewer_nxgA · 2025-07-03

**Clarity:** 4
**Significance:** 3
**Originality:** 3
**Rating:** 5
**Confidence:** 4

**Summary:**

- This work proposes Disentangled Action and Context concept-based Explainable (DANCE) video action recognition

- The DANCE framework uses a frozen video backbone to extract features, then projects these features into a set of interpretable object, scene, and motion dynamics activations

- These interpretable concept activations are used as features for a linear action prediction layer

- To obtain a set of discrete motion dynamics concepts, they extract 2D pose sequences with a pose estimation model from the training videos and cluster the pose sequences. A video’s poses are then represented as a binary multi-label vector indicating which clusters the video’s pose sequences are members of

- To obtain object and scene concepts, they prompt an LLM to list the likely objects and scenes associated with each action class

- Training proceeds by first training the projection layers to predict the motion dynamics, then training a linear layer on the predicted concepts

- They find that DANCE’s explanations are heavily preferred to other interpretable baselines, and that classes are readily interpretable based on highly-weighted concepts

- The model’s interpretability allows for straightforward model editing, and they show that this process can improve classifications under distribution shift

**Questions:**

Please see the weaknesses for changes I would like to see addressed (providing more context of DANCE's performance relative to baselines).

For clarification: It is unclear what “when this concept is deactivated” means on line 314. Is it assigned a weight of zero?

**Ethical Concerns:**

["NO or VERY MINOR ethics concerns only"]

**Final Justification:**

The authors have addressed my concerns regarding
1. Providing a comparison to non-interpretable baselines
2. Potential inflexibility of the framework by providing a an example of how one would avoid full retraining
and have clarified a point of confusion.

The paper presents a well-motivated, high-performance method for interpretable video action recognition. The paper is lucid and, with the additional discussions/experiments, the well-rounded. Therefore, I maintain my recommendation of 5 for acceptance.

**Limitations:**

Yes

**Paper Formatting Concerns:**

No concerns.

**Quality:**

3

**Strengths And Weaknesses:**

# Strengths

- Straightforward, effective approach to an intrinsically interpretable action recognition model

- Results show a clear preference for their model over saliency-based methods and concept bottleneck models with spatio-temporal concepts, showing the benefit of pose-based interpretable features

- The writing is clear and the figures informative

- Qualitative examples are convincing of the approach’s explainability

- Manual model editing to avoid complete model retraining is a useful capability of this approach


# Weaknesses

- The non-interpretable baseline in Table 1/line 296 is not described. How comparable is this model? Is it finetuned, zero-shot? [PapersWithCode](https://paperswithcode.com/sota/action-recognition-in-videos-on-ucf101) suggests that the top performance on UCF101 is 99.70, for example—significantly higher than the performance of the baseline w/o interpretability’s 88.4%. It is reasonable to focus on appropriate comparisons to DANCE, but more context for the baseline and SOTA models should be provided

- This method trades off flexibility for the sake of interpretability—adding a new class would likely require retraining the entire pipeline (e.g. mining concepts and clustering from scratch). A flexible but opaque alternative would simply be to use an open-vocabulary model. An explicit comparison with open-vocabulary, or more readily extensible alternatives, would strengthen the evaluations

---

> ### Author Rebuttal · Authors · 2025-07-29
>
> We thank the reviewer for the helpful comment.
> ### **Clarification on the baseline in Table 1**
> We apologize for the lack of clarity regarding the "baseline w/o interpretability" and other baselines in Table 1. The "baseline w/o interpretability" refers to a standard video recognition model trained without the linear concept layer. Specifically, we fine-tune a video encoder, e.g., VideoMAE, and a linear classification head directly on each target dataset without any concept supervision. We use Somethine-Something-v2-pretrained VideoMAE weights for KTH, Penn Action, and HAA-100, and UCF101-pretrained weights for UCF-101 to match the pretraining setup used in DANCE. Since DANCE adopts VideoMAE as its video backbone fine-tuned on each target dataset, we report the Top-1 accuracy of VideoMAE across datasets as the “baseline w/o interpretability” in Table 1. Also, for a fair comparison, isolating the effect of interpretability, we use the same feature extractor (VideoMAE) across all baselines (e.g., CBM [1] with UCF-101 attributes, LF-CBM [2] with entangled language concepts, and LF-CBM [2] with disentangled language concepts). This follows the common practice in ante-hoc explainable models [1,2,5,6], where the baseline is typically defined as the same backbone architecture trained for the target task without the interpretability. While VideoMAE is used throughout Table 1 of the main paper for consistency, the backbone is interchangeable, and we additionally report results with other backbones (e.g., CNN) in Table 10 of the supplementary material.
>
> ### **Justification for SOTA Comparison**
> We agree that the performance of DANCE is lower than recent SOTA results on UCF-101 (e.g., 99.7%). However, our goal is not to compete with SOTA in recognition accuracy. Instead, our focus lies in injecting interpretability into video backbones and analyzing the trade-off between interpretability and performance in Table 1 of the main paper.
> Nevertheless, to examine the scalability of DANCE and its compatibility with stronger architectures, we apply it to AIM [3] and InternVideo2 [4]—both of which are more powerful backbones than VideoMAE.  Specifically, AIM is fine-tuned on UCF-101 (AIM is a parameter-efficient fine-tuning method using a pre-trained CLIP as its backbone), while InternVideo2 is used directly with its IV-400M pretraining.  We present the results in the table below.
> | Method         | Interpretability | Top-1 Accuracy |
> |----------------|:------------------:|:-----------------------------------:|
> | AIM           | ✗                | 94.5                       |
> | DANCE with AIM backbone         | ✓                 | 93.6                       |
> | Internvideo2 ( visual )    | ✗                | 97.3                       |
> | DANCE with InternVideo2 backbone ( visual )      | ✓                | 96.5                       |
>
> AIM alone achieves 94.5% on UCF-101, and DANCE with AIM backbone achieves 93.6%. Similarly, InternVideo2 achieves 97.3% alone, and DANCE with InternVideo2 backbone achieves 96.5%. Specifically, we use only the frozen visual encoder of InternVideo2 as our video feature extractor. These results show that DANCE consistently benefits from stronger backbones while maintaining competitive performance.
>
> ### **More context on DANCE’s performance relative to baselines**
> We provide additional context on how DANCE performs relative to baselines. Because DANCE enforces the video encoder to predict concepts through a linear concept layer, there might be a trade-off between accuracy and interpretability [1]. In this setting, it becomes crucial to assess how well each method maintains recognition performance while providing meaningful explanations. As shown in Table 1 of the main paper, DANCE achieves higher accuracy than other concept-based methods on KTH and Penn Action, while showing only a modest drop of 2.8 points on HAA-100 and 0.9 points on UCF-101 compared to the baseline w/o interpretability. Furthermore, as shown in Figure 6 of the main paper, DANCE receives the highest user ratings in the user study evaluating explanation quality. These results demonstrate that DANCE achieves the best trade-off between interpretability and performance, validating the effectiveness of our explanation design.
>
>
> ### **Flexibility-interpretability trade off**
> We acknowledge that DANCE prioritizes interpretability, which may impose some constraints on flexibility. However, adding a new class does not require redoing concept discovery from scratch.
>
> For object and scene concepts, we can extract concepts from the new class and apply concept pseudo-labeling for the new concepts and the samples belonging to the new class (see Eq. (3) in the main paper). For motion dynamics concepts, we can discover new clusters from the new class’s pose sequences using a semi-supervised clustering approach [7], allowing us to incrementally expand the concept space when novel motion patterns emerge. Therefore, we can add a new class by updating concept labels and retraining only the concept layers and the linear classifier. Retraining takes only $\sim 8$ minutes with a single RTX 3090 GPU on the UCF-101 dataset.
>
> To empirically verify this, we simulate a class addition scenario on the UCF-101 dataset by randomly holding out one class during training. After training DANCE on the 100 classes, we update the object, scene, and motion dynamics concept labels for the held-out class without concept discovery from scratch. We show the results of this scenario in the table below.
>
> | Method| Top-1 Accuracy |
> |---------------------|:---------:|
> | DANCE before adding a new class |   87.2    |
> | DANCE after adding a new class  |   86.8    |
>
> The marginal accuracy drop (< 0.5 points) confirms that DANCE supports efficient class extension without re-running the full concept discovery pipeline. While fully incremental learning (i.e., class-incremental learning with many classes) is outside the scope of this work, we believe this experiment demonstrates that DANCE is more extensible than a rigid, from-scratch formulation.
>
> ### **Comparison with open-vocabulary methods**
> To address the reviewer’s concern, we compare the performance of DANCE with recent open-vocabulary models on the UCF-101 dataset. We present the results in the table below.
> | Method   | Total Param.     | Interpretability | Top-1 Accuracy |
> |------------|:----:|:------------------:|:----------------------------:|
> | CLIP [8]          |150M| ✗                | 68.9                       |
> | TVTSv2 [9]      | 956M | ✗                | 78.0                       |
> | InternVideo2 [4]| 6B | ✗                | 89.5                       |
> | **DANCE**    | 90M  | ✓                | 87.5                       |
>
> While InternVideo2 shows the highest accuracy (89.5%), it is a 6B-scale foundation model with multi-modal capabilities. These results suggest that DANCE achieves a favorable balance between interpretability and performance, remaining competitive even when compared to large-scale open-vocabulary methods.
>
> ### **Clarification on concept deactivation**
> We apologize for the lack of clarity. In this context, “deactivating” a concept means setting its activation to zero, not its weight. This allows us to assess the influence of each concept by directly manipulating the output of the concept layer while keeping the learned weights unchanged. We will clarify this point explicitly in the final version.
>
>
> ##### **References**
> [1] Koh, Pang Wei, et al. "Concept bottleneck models." ICML 2020.
>
> [2] Oikarinen, Tuomas, et al. "Label-free Concept Bottleneck Models." ICLR 2023.
>
> [3] Yang, Taojiannan, et al. "AIM: Adapting Image Models for Efficient Video Action Recognition." ICLR 2023.
>
> [4] Wang, Yi, et al. "Internvideo2: Scaling foundation models for multimodal video understanding." ECCV 2024.
>
> [5] Yuksekgonul, Mert, Maggie Wang, and James Zou. "Post-hoc Concept Bottleneck Models." ICLR 2023.
>
> [6] Shang, Chenming, et al. "Incremental residual concept bottleneck models." CVPR 2024.
>
> [7] Vaze, Sagar, et al. "Generalized category discovery." CVPR. 2022.
>
> [8] Radford, Alec, et al. "Learning transferable visual models from natural language supervision." ICML 2021.
>
> [9] Zeng, Ziyun, et al. "Tvtsv2: Learning out-of-the-box spatiotemporal visual representations at scale." arXiv preprint arXiv:2305.14173 (2023).

---

> > ### Comment · Reviewer_nxgA · 2025-08-05
> >
> > Thank you for your response and the additional experiments. The provided points are clear.

---

### Note · Authors · 2025-08-13

We sincerely thank all reviewers for their constructive feedback. DANCE is a novel XAI framework for video action recognition that explicitly *structures* the model to reason through three disentangled, human-interpretable concept spaces—motion dynamics, objects, and scenes—discovered entirely in a label-free manner. Prior XAI methods in video action recognition relying on saliency maps, language descriptions, or expert-defined attributes struggle to capture *fine-grained temporal cues*. We address this gap by introducing *motion dynamics concepts* derived from skeleton sequences, which disentangle temporal patterns from spatial context and enable more faithful and human-aligned explanations of the model’s predictions.

During the rebuttal period, we have provided additional analyses and clarifications that address all reviewer concerns. Specifically, we have strengthened the comparison with language-based explanation baselines, showing that our structured concept design has overcome the limitations in capturing temporal cues and disentangling spatial and temporal factors. We have also reinforced the clustering analysis of motion dynamics concepts with quantitative evidence and a user study, confirming that the discovered concepts are coherent and interpretable. In addition, we have addressed suggestions for future directions, including multi-person scenarios and non-human action recognition, while providing clarifications or evidence for all the other comments.

We kindly ask the ACs and the reviewers to review the demo video included in the supplementary material, which clearly illustrates how DANCE has generated intuitive, human-aligned explanations for an input video. These visual demonstrations make the interpretability tangible and highlight how our framework delivers disentangled and structured explanations that are faithful to the model’s decision process. We believe DANCE marks a significant step toward building trustworthy and interpretable XAI for video action recognition. Once again, we thank the reviewers for their constructive engagement, which has further strengthened the quality of our work.

---

### Decision · Program_Chairs · 2025-09-17

**Decision:**

Accept (spotlight)

**Comment:**

This work received mostly positive reviews in the initial review, with 2 borderline accept and 2 accept scores. While the work was generally seen as technically sound and timely, some concerns raised around the following key points: (i) ambiguous technical novelty that does not clarify how it moves beyond the standard concept‑bottleneck pipeline and modest performance gains, (ii) potential over-dependence on precomputed, dataset‑specific motion clusters and GPT‑generated object/scene lists can limit its generalization to new domains or unseen actions, and (iii) limited details on baselines and limited qualitative analysis such as robustness checks and cluster quality analysis to support interpretability and effectiveness claims.

The authors have provided a rebuttal that addresses many of these concerns effectively, as seen by the increase in scores to 3 accepts and 1 borderline accept. The AC agrees with the general assessment and recommends acceptance. It is a good paper that will generate a lot of discussion with the NeurIPS audience. The authors are strongly encouraged to incorporate the discussion from the rebuttal phase into the final, camera-ready version for completeness.